psychology/cognition

consolidation, word learning, encoding, psycholinguistics, word recognition, lexicon

**Author for correspondence:**
M. G. Gaskell
e-mail: gareth.gaskell@york.ac.uk

# Learning to live with interfering neighbours: the influence of time of learning and level of encoding on word learning

S. Walker, L. M. Henderson, F. E. Fletcher, V. C. P. Knowland, S. A. Cairney and M. G. Gaskell

Department of Psychology, University of York, York YO10 5DD, UK

SW, 0000-0002-5613-5428

New vocabulary is consolidated offline, particularly during sleep; however, the parameters that influence consolidation remain unclear. Two experiments investigated effects of exposure level and delay between learning and sleep on adults' consolidation of novel competitors (e.g. BANARA) to existing words (e.g. BANANA). Participants made speeded semantic decisions (i.e. a forced choice: natural versus man-made) to the existing words, with the expectation that novel word learning would inhibit responses due to lexical competition. This competition was observed, particularly when assessed after sleep, for both standard and high exposure levels (10 and 20 exposures per word; Experiment 1). Using a lower exposure level (five exposures; Experiment 2), no post-sleep enhancement of competition was observed, despite evidence of consolidation when explicit knowledge of novel word memory was tested. Thus, when encoding is relatively weak, consolidation-related lexical integration is particularly compromised. There was no evidence that going to bed soon after learning is advantageous for overnight consolidation; however, there was some preliminary suggestion that longer gaps between learning and bed-onset were associated with better explicit memory of novel words one week later, but only at higher levels of exposure. These findings suggest that while lexical integration can occur overnight, weaker lexical traces may not be able to access overnight integration processes in the sleeping brain. Furthermore, the finding that longer-term explicit memory of stronger (but not weaker) traces benefit from periods of wake following learning deserves examination in future research.

# 1. Introduction

An accumulation of behavioural and neuroimaging research now suggests that sleep is one state that actively supports the consolidation of a newly encountered word, and specifically, the extent to which it becomes *integrated* with existing lexical knowledge [1–4]. Such findings can be explained by complementary learning systems (CLS) accounts (e.g. [5,6]), which rest on the assumption that periods of consolidation are required for hippocampal memory traces to be reactivated and reach long-term neocortical systems. Despite this overnight emergence of lexical integration being commonly reported at the whole group level, the overnight changes are variable and sensitive to individual differences [7–9] and training and testing environments [10–13]. A systematic evaluation of the parameters that influence overnight lexical integration is therefore warranted.

## 1.1. Lexical integration

Once a new word has been successfully integrated into the lexicon, it should behave as an existing word and compete with its lexical neighbours for recognition. In a series of studies by Gaskell and colleagues [2,3,14], adult participants were taught novel spoken items (e.g. CATHEDRUKE) that overlapped phonemically with existing words (e.g. CATHEDRAL). Engagement of the novel words in lexical competition was tested by asking participants to make speeded decisions about the presence or the absence of a pause inserted near the offset of the existing word. It was hypothesized that if CATHEDRUKE became lexically integrated, then the uniqueness point of CATHEDRAL (the point in the left-to-right phonemic sequence at which it diverges from other words) should shift towards its offset, thus increasing lexical activity at the pause and delaying pause detection [14]. The findings indicated that lexical competition effects did emerge, but typically only after a delay including sleep [3,14,15]. Furthermore, Dumay & Gaskell [3] administered a free recall task both immediately after training and 12 h later: participants were significantly better at recalling the novel words at the 12 h test, but only if they had slept. Together, these results fit with computational theories that argue for a dual system of word learning to guard against 'catastrophic interference' (e.g. the CLS account; [6,16]). Such theories propose that novel word forms are initially stored using hippocampal mediation, allowing for rapid learning of new forms and linking to appropriate lexical knowledge. In order to fully integrate this new mapping into the lexicon, the mediating role of the hippocampus is transferred to the neocortex, via a more enduring process that may require sleep [17,18]. It should be acknowledged, however, that sleep-associated delays in the emergence of lexical competition may depend upon the training conditions and the test of lexical competition that is used [11–13].

A smaller but parallel literature has studied the acquisition and integration of written words. Bowers *et al.* [19] trained adults on a series of novel words (e.g. BANARA) that were derived from existing words (e.g. BANANA). As all existing words were 'hermit' words (i.e. words with no orthographic neighbours derivable via the substitution, deletion, transposition or addition of a single letter), it was thought that the integration of the novel word into the lexicon should have the effect of altering the neighbourhood size of the existing word from zero to one. Using a semantic categorization task, where speeded decisions were made to the existing word (e.g. is the word a natural or artefact item), findings were akin to the effects seen with pause detection tasks in spoken word learning studies. That is, response times for accurate responses on the semantic categorization task were significantly slower to the words with a new neighbour in comparison with control hermit words without a newly learned neighbour. This effect was observed on the day after training, but not immediately after training, again suggesting a role of offline consolidation in orthographic lexical integration (although cf. [20] for an alternative account). Wang *et al.* [21] further showed that the orthographic lexical competition effect was evident after a period of sleep (i.e. for participants trained in the evening and tested the next morning), but not after an equivalent period of wake (for participants trained in the morning and tested the same evening). Just like the spoken word literature, though, there are conditions in which sleep-associated enhancements have not been documented, such as when encoding involves 'fast mapping' [10].

## 1.2. Understanding individual differences in lexical integration

An important next step in this line of research is to determine the factors that influence the course of lexical integration and the conditions in which sleep is important. This study focused on the impact and interaction of two key variables—the delay between learning and sleep and the level of encoding.

A large body of research suggests that what ensues in the minutes to hours that immediately follow new encoding has an impact on the retention of that new information [22–26]. Preliminary steps have also been made to study the influence of the simple passing of time between learning and sleep on declarative memory consolidation (e.g. paired-associate learning). This research suggests that newly encoded memories are subject to interference and forgetting during wake [27] and that sleeping shortly after learning has a pronounced benefit relative to time spent awake for both verbal [28,29] and non-verbal declarative memory [30].

The conclusion that sleeping soon after learning always aids retention has been challenged by one study showing that a delay between learning and sleep (i.e. 4 h), in comparison to immediate sleep or wakefulness, can *enhance* declarative memory retention in adults [31]. Therefore, in some cases, it may be possible that a period of wakefulness allows for additional processing in order to strengthen the memory trace to an optimal level for sleep-related effects (consistent with claims that a certain level of performance is required for sleep effects to emerge; [32]). Furthermore, significant delays between learning and sleep often occur in the sleep conditions of typical sleep–wake experiments (e.g. 3–4 h in Gais *et al.* [28], which in fact is comparable to the 'delayed' condition in Alger *et al.* [31]), and often the precise delay is not reported. Thus, a more systematic evaluation of the effect of time between encoding and subsequent sleep on memory consolidation is needed. To this end, this study compared the effects of time between learning and nocturnal sleep on both the explicit retention and the lexical integration of visually presented novel words.

We also explored the influence of encoding on overnight lexical integration. It has been shown that improved performance on a motor learning task (adaptation to a systematic directional error on cursor movement) not only correlates with an increase in slow-wave activity (SWA) during intervening sleep, but that this increase in SWA is correlated with enhanced performance following sleep ([33], for comparable findings in mice and rats, see [34,35]). Similarly, using three different measures of declarative memory, Tucker & Fishbein [36] showed that adults who napped following training showed clear sleep-dependent performance benefits compared to non-napping controls; however, this was only the case for adults who performed in the top half of the sample during training. It was argued that 'subjects that demonstrate greater facility to learn each of the tasks are better equipped physiologically to benefit from sleep-related mnemonic processes'. It should be acknowledged, however, that other studies have found that sleep consolidation benefits are greater at intermediate [32] or even *weaker* levels of performance [27,37,38]. Despite these inconsistencies (which are probably a consequence of specific task demands), the data lead to the prediction that an individual's encoding experience is likely to be an important predictor of how well sleep facilitates memory making.

In manipulating encoding *and* the learning–sleep interval, this study was set up to examine a potential interaction between these two variables. High levels of exposure during learning have been found to lead to subsequent lexical competition effects despite a long interval between learning and sleep [14]. Thus, an interaction between encoding strength and the learning–sleep interval was predicted, such that items encoded to a higher level may give rise to stronger memory traces that are less susceptible to interference during wake.

Two experiments were carried out to investigate the effects of encoding and the learning–sleep interval on orthographic lexical integration. Lexical competition with neighbouring words (as measured via a semantic categorization task, following Bowers *et al.* [19]) was used as a marker of lexical integration, whereas recognition and recall tests on the novel words were used as measures of explicit retention. Both experiments incorporated the time between learning and sleep as a continuous predictor, with Experiment 1 training novel words at standard [19,21] and high levels of exposure and Experiment 2 using a lower level of exposure. An online, web-based data collection procedure was employed via Qualtrics (Qualtrics, Provo, UT). Unlike previous laboratory-style tests, this procedure allowed a naturalistic manipulation of time delay which spanned up until participants' typical bedtimes (cf. [39]).

## 2. Experiment 1

Experiment 1 (pre-registered at osf.io/zntvp) adopted an encoding procedure known to show sleep-related benefits in the lexical integration of pseudowords (e.g. BANARA; [21,22]), and using a within-subjects design, compared this 'standard' level of exposure to a more intensive level of exposure. We tested participants' memory for these items immediately after learning and again the following day. In line with previous studies, we predicted an increase in the strength of lexical competition and improvements in explicit memory after a period of offline consolidation that included sleep. The delay

between learning and bedtime onset was manipulated to test the hypothesis that a shorter delay would lead to greater overnight improvements in lexical competition and explicit memory (although note the reverse effect could be predicted based on Alger *et al*. [31]). This effect was predicted to be moderated by the level of encoding, such that words trained in the more intensive encoding condition would demonstrate more robust consolidation effects and be less susceptible to a delay between learning and bedtime.

## 2.1. Method

### 2.1.1. Participants

Our pre-registration specified 80 participants. In order to meet this target with usable datasets, 113 monolingual English speakers with no known reading, language, developmental or psychological disorders were recruited. Participants were students at the University of York aged 18–33 years (mean = 20.09 years, s.d. = 2.28 years; 16 males) and were paid or received course credits.

Participants took part in two encoding conditions over two consecutive weeks (referred to as Week 1 and Week 2). During both weeks, as pre-registered, they were asked to refrain from caffeine, alcohol and cigarettes on the day of training and the morning of testing, as is standard for this type of experiment. Non-compliance was determined as non-attendance, incomplete attendance, completing the tasks at the wrong time of the day, admitting to drinking caffeine or alcohol, or admitting to smoking. Twenty participants did not comply with at least one of these instructions in Week 1, and 29 in Week 2. Rather than eliminating their data altogether, their data for the week(s) of non-compliance were removed, leaving the numbers remaining (Week 1 N = 93; Week 2 N = 84) close to the pre-registered target of 80.

### 2.1.2. Materials

*Stimuli.* Eighty hermit words that had no single-letter orthographic neighbours (according to [40]) were selected (e.g. BANANA) as 'base' words for the purposes of creating pseudoword competitors (32 of which were taken from Bowers *et al*. [19]). All base words were concrete nouns, ranging in length from five to seven letters, with a CELEX per-million frequency of 3–38 [41]. Half (40) of the base words were naturally occurring items (e.g. BANANA) and half were man-made items (e.g. ANCHOR) for the purposes of the semantic categorization task (see below). Novel pseudowords were constructed by substituting one internal letter of each base word to form a pronounceable non-word (e.g. BANARA) [19]. An additional set of 40 words (ranging in length from five to seven letters) was created to be used as fillers in the semantic categorization task. Each filler word was a concrete noun with a CELEX frequency ranging between 1 and 492 counts per million. The fillers were a mixture of hermit and non-hermit words. Two sets of 40 critical words were used for each encoding condition (administered in two separate, consecutive weeks). For each set of 40 words, each critical word gained a neighbour for half the participants and remained a hermit word for the other half. Thus, for each encoding condition, participants learned 20 novel words (10 from the natural category and 10 from the man-made category). In total, four matched, counterbalanced lists of 20 novel words were created. In addition, for the speeded recognition task (described below), 40 non-word foils were used. These foils were generated from five, six or seven letter nouns with a CELEX frequency of 2–710 per million. Similar to the novel words described above, the non-word foils were devised by substituting one internal letter of the noun to form a pronounceable non-word (e.g. critical word: TICKET; foil: TILKET).

*Sleep measures.* The Pittsburgh sleep quality index (PSQI) [42] assessed general sleep quality over the month preceding participation. This questionnaire is made up of 19 self-rated items, which are used to form seven component scores: subjective sleep quality, sleep latency, sleep duration, habitual sleep efficiency, sleep disturbances, use of sleeping medication and daytime dysfunction. The sum of scores for these seven components yields one global score with a maximum of 21 (mean PSQI global score for the standard exposure condition = 5.13, s.d. = 2.57, 1–16; mean PSQI global score for the high exposure condition = 5.30, s.d. = 2.65, 1–16).

### 2.1.3. Procedure and design

Exposure level was manipulated within participants, with participants completing both standard and high exposures, spaced one week apart. In each case, participants were tested immediately after exposure as well as the following morning (Day 1 versus Day 2). As shown in figure 1, order of

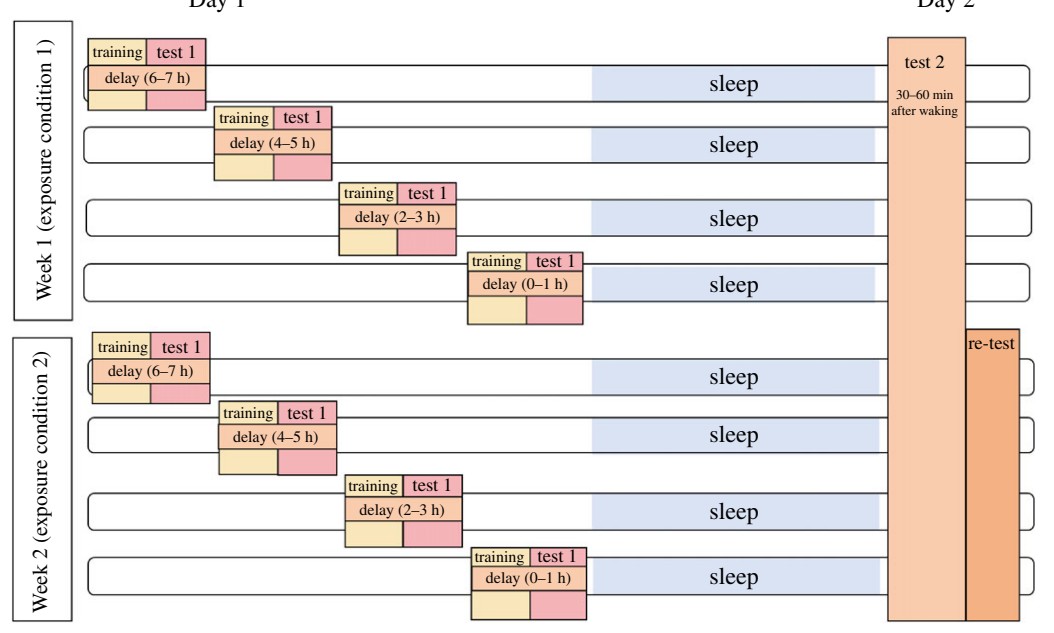

**Figure 1.** Experimental procedure. In Week 1, participants were trained on the novel words and immediately tested in accordance with their delay allocation (0–1 h, 2–3 h, 4–5 h or 6–7 h), before being tested again the following morning. In Week 2, the same participants completed the alternative Exposure condition (and were instructed to follow the same delay allocation), before being re-tested on items from Week 1.

exposure to the standard and high sets was counterbalanced across participants. Participants were allocated randomly to one of four groups, which determined their time interval between learning and bedtime (0–1 h, 2–3 h, 4–5 h, 6–7 h) for both of their exposure sessions. This was done to elicit a wide spread of intervals between exposure and sleep and to allow us to encapsulate the learning–sleep interval (*Delay*) as a continuous variable. The learning–sleep interval was defined as the time in minutes between the end of the training session on Day 1 (determined electronically via the online experiment package, Qualtrics; Qualtrics, Provo, UT) and the participant's self-reported bedtime on Day 1 (recorded by the participant via Qualtrics on Day 2). For the semantic categorization task, there was an additional within-participants independent variable, Word Type (hermit versus non-hermit). For the hermit items, the neighbouring pseudoword was not learned and so these acted as a control. For the non-hermit items, the neighbouring pseudoword was included in the set of words to be learned, potentially changing the competition environment of the existing word.

All testing sessions were completed online via Qualtrics and QRTEngine [43]. Participants were asked to carry out the experiment in a quiet environment, free from distraction. On Day 1, participants were asked to complete a training phase (described below). Depending on their delay group, participants received instructions that determined the approximate time to start training (i.e. they were asked to complete the training roughly 0–1 h, 2–3 h, 4–5 h or 6–7 h before bedtime). Test time was checked and recorded for each participant. As mentioned, we used Delay as a continuous variable in our analyses (mean Delay Standard exposure = 203.51 min, s.d. = 139.02 min, 0–525 min; mean Delay High exposure = 204.20 min, s.d. = 138.05 min, 0–483 min). Immediately following training, participants completed a testing phase, which consisted of a psychomotor vigilance test (PVT; to allow us to determine whether any potential effects of Delay could be attributed to time of day/fatigue), a semantic categorization task (to measure lexical integration of the newly learned pseudowords), and free recall and speeded recognition tasks (to measure explicit memory of the novel items). On Day 2, approximately 30–60 min after waking, participants completed all testing phase tasks again, in the same order as Day 1. Following Day 2 testing, participants were asked about the activities they took part in between training/testing on Day 1 and bedtime on Day 1, to allow us to exclude participants who engaged in activities that might have influenced nocturnal sleep (i.e. napping, drinking alcohol or drinking caffeine). The PSQI and a series of standard sleep questions regarding bed, sleep and wake time were also completed. This procedure was repeated approximately one week later (minus the PSQI), when participants completed the alternate Exposure condition. In Week 2, all participants were given the same instructions as in Week 1 regarding the approximate time they should complete training prior to

bedtime. Once testing was complete on Day 2 of Week 2, participants were re-tested on the words learned from Week 1 to explore the maintenance of any effects over a longer time frame.

*Training*. Novel words were presented on a computer screen via Qualtrics in upper case 28 point Arial font. Each novel word was presented simultaneously above a text box, and participants were asked to type out the word as accurately as possible. Following Bowers *et al*. [19], participants were informed of incorrect responses, and asked to type the word again until correct. Once typed correctly, participants were able to click a button to move on to the next word. In the standard exposure condition, we matched our exposure level to Bowers *et al*. [19], with each novel word typed 10 times (total of 10 blocks). In the high exposure condition, the exposure level was doubled (20 blocks). For each participant, the novel words were presented in a randomized order within each block. As in Bowers *et al*. [19], at the end of each block, a full list of the novel words was presented, and the participants were asked to carefully read through the list.

*Testing*. Participants were asked to complete a vigilance task, a semantic categorization task, a free recall task and a speeded recognition task during testing.

*Vigilance*. Prior to completion of the main experimental tasks, participants completed a PVT (based on Basner & Dinges [44]). Participants were presented with a cross on the screen after varying interstimulus intervals (ISIs) and told to respond as quickly as possible by pressing the spacebar. There were 24 trials in total. Timeouts were set to 1000 ms. ISIs were randomly sampled for each participant from a flat distribution ranging from 2000 to 8000 ms. A series of partial correlations were conducted to assess the impact of vigilance at encoding on the relationship between Delay and overnight change in (i) the competition effect (semantic categorization task), (ii) reaction time (speeded recognition task) and (iii) accuracy (recall task). The effect of vigilance was explored using both mean reaction time and percentage of hits to the vigilance task. No significant correlations were found.

*Semantic categorization*. Participants were asked to classify, as quickly and accurately as possible, each of the base words and fillers as members of the natural or man-made categories by pressing the Z or M keys on a computer keyboard, respectively. Eighty items were randomly presented during each testing phase (40 base words and 40 fillers). Each trial began with a fixation cross displayed for 500 ms, followed by the target word presented in upper case 28 point Arial font, which was displayed until the participant responded or a 3000 ms timeout was reached. Ten practice trials preceded the 80 experimental trials. Feedback was provided during the practice trials; but not during the experimental trials.

*Free recall*. Participants were asked to type out as many of the novel words as they could remember. Participants were told that there were 20 words in total. No time limit was given for this task.

*Speeded recognition*. Participants were presented with the newly learned items intermixed with the foils. They were asked to decide as quickly and accurately as possible if they had or had not learnt the presented word by using the Z and M keys on the keyboard, respectively. There were 40 trials in total (20 novel words and 20 foils). The trial structure (fixation cross, trial times, font type and size) was the same as that used in the semantic categorization task.

### 2.1.4. Analysis

Although we had pre-registered an analysis using maximal random effects structures, there is now increasing awareness of the potential for maximal models to reduce power due to unnecessary complexity [45]. Therefore, we used a parsimonious approach in line with Bates *et al*. [46].

Data were analysed using R [47], with models fitted using the package lme4 [48] and figures made using ggplot2 [49]. Logistic mixed-effects regression models were used to model binary outcomes (free recall accuracy) and linear mixed-effects models for continuous outcome data (semantic categorization and speeded recognition reaction times (RT)). Only correct responses were analysed for RT data. For each dependent variable, a mixed model was fitted, with fixed effects (i.e. all main effects and interactions) of *Exposure* (standard versus high), *Word Type* (hermit versus non-hermit),[1] *Day* (day 1 versus day 2) and *Delay*. Delay was centred and scaled by subtracting the mean from each value and dividing by the standard deviation in order to enhance the interpretation of the beta coefficient and to normalize the spread of scores [50,51]. Categorical predictors were coded using deviation coding to assess each main effect and interaction independently of other predictors in the model (Day: day 1 = −0.5, day 2 = 0.5; Exposure condition: standard = −0.5, high = 0.5; Word Type: non-hermit = −0.5, hermit = 0.5). Based on tests of normality, an inverse transformation (−1000/RT) was used to normalize the distribution of RTs [52].

---

[1]Word Type was only included in the model for the semantic categorization task.

We followed the recommendations of Bates *et al*. [46] for specifying the best-fitting maximal model for each dependent variable. This procedure involved (i) determining the best-fitting fixed effects structure from an intercept-only model, (ii) justifying the random intercepts, and (iii) establishing random slopes for each justified random intercept.

To determine the best-fitting fixed effects structure, a model with a maximal fixed effects structure and random intercepts only was used. A backwards selection procedure was used [46]. That is, each interaction within the fixed effects structure was removed one at a time, with the highest order interactions explored first. At each stage, the model was compared to each previous model, using likelihood ratio tests to determine any model change using $p < 0.2$. The threshold of $p < 0.2$ was used to guard against anti-conservativity, in line with the recommendations of Barr *et al*. [53]. Where the removal of a fixed effect did not affect the model (i.e. $p > 0.2$), the removal of this fixed effect was deemed justifiable. In addition, where the removal of a fixed effect was not justified, all lower order interactions were retained (see osf.io/zntvp). This procedure was repeated until all fixed effects were analysed and the final fixed effects structure was determined.

To justify the random intercepts, each intercept was removed separately from the final fixed effects structure and compared to the final fixed effects structure containing all intercepts, using likelihood ratio tests. As above, a threshold of $p < 0.2$ was used to indicate justification for model inclusion. For all analyses within this experiment, there were two random intercepts present; Participant and Item. The intercept that contributed most when compared to the final fixed effects model was explored first when establishing random slopes.

A forward model selection process was used to determine random slopes. Only those main and interaction effects present in the fixed effects structure were explored. Each main and interaction random slope was added to the intercept one at a time (again using a criterion of $p < 0.2$). Random slopes for main effects were established first. The model with the lowest $p$-value was selected and compared against when establishing interaction slopes. This was repeated until no further improvement ($p < 0.2$) could be achieved.

The best-fitting models are interpreted as a standard regression model; a positive *b* coefficient means that the independent variable showed a positive relationship with the dependant variable. The $p$-values were provided by lmerTest [54].

*Outlier removal.* Accuracy rates were averaged across all days for the semantic categorization task. Participants with accuracy rates 2.5 standard deviations below the group mean were excluded ($n = 1$). For the speeded recognition task, in 14/342 sessions of data, accuracy was substantially and significantly *below* chance (mean 6% accuracy, with chance at 50%). Since this task was completed outside of the laboratory, and a reminder of the response keys was not displayed during trials, it was assumed that these participants had mistakenly reversed the response key associations. Subsequently, we re-reversed key assignment in recording their correct and incorrect responses, on a session-by-session basis. A d-prime calculation was then used to determine participant outliers separately for each week. Two participants were identified as extreme low outliers in the d-prime distribution for Week 2 (d-prime cutoff 1.5) and subsequently excluded from the Week 2 analysis. No participants were excluded for response bias in Week 1.

For both the semantic categorization and speeded recognition tasks, within-subjects outliers were classed as any trials 2.5 s.d. below a participants' mean RT. For individual items, accuracy rates across all days were averaged separately for the semantic categorization and speeded recognition tasks. Items with accuracy rates 2.5 s.d. below the group mean were excluded (i.e. CHALK was removed from the semantic categorization task and SPIMER and ULPER were removed from the speeded recognition task).

*Qualtrics check.* As the experiment was conducted online using QRTEngine, it is advised to check for server communication delays which could contaminate RT data [43]. In order to analyse this, for each RT task, the time it takes for a trial screen to disappear was averaged for each participant. If this mean $+ 2$ s.d. exceeded 2000 ms, the participant was removed from the RT analysis [43]. In the semantic categorization task, 12 participants were identified as having issues with their server communication. Four of these were removed from the Week 1 analysis, and eight from Week 2. In the speeded recognition task, six participants were identified as having issues with their server communication (four from Week 1 and two from Week 2).

## 2.2. Results

Descriptive statistics for all main study variables can be found in table 1.

**Table 1.** Experiment 1 mean (and s.d.) RTs for the semantic categorization and speeded recognition tasks (ms) and free recall accuracy (%).

| exposure | day | semantic categorization | | | speeded recognition | free recall |
|---|---|---|---|---|---|---|
| | | hermit | non-hermit | difference | | |
| standard | 1 | 888 (167) | 926 (177) | 38 (97) | 755 (184) | 45.5 (24.0) |
| | 2 | 845 (169) | 926 (257) | 82 (154) | 740 (153) | 49.2 (23.5) |
| high | 1 | 887 (195) | 925 (216) | 38 (114) | 735 (150) | 45.3 (23.0) |
| | 2 | 836 (161) | 895 (193) | 59 (114) | 751 (166) | 48.3 (22.6) |

**Table 2.** Experiment 1 predictors of semantic categorization RT performance. Italics denote $p < 0.05$.

| fixed effects | b | s.e. | t | p |
|---|---|---|---|---|
| *(Intercept)* | *−1.24* | *0.02* | *−64.81* | *<0.001* |
| *Word Type* | *0.05* | *0.01* | *7.30* | *<0.001* |
| *Delay* | *0.04* | *0.02* | *2.48* | *0.01* |
| Exposure | −0.02 | 0.01 | −1.32 | 0.19 |
| *Day* | *−0.05* | *0.01* | *−4.55* | *<0.001* |
| Word Type : Delay | 0.01 | 0.01 | 1.11 | 0.27 |
| Delay : Exposure | 0.001 | 0.02 | 0.09 | 0.93 |
| *Word Type : Day* | *0.03* | *0.01* | *2.58* | *0.01* |
| Delay : Day | 0.01 | 0.01 | 0.06 | 0.54 |
| Exposure : Day | −0.002 | 0.02 | −0.11 | 0.92 |
| *Delay : Exposure : Day* | *−0.04* | *0.02* | *−2.37* | *0.02* |
| random effects | variance | s.d. | | |
| Participant: (intercept) | 0.03 | 0.16 | | |
| Participant: Day (slope) | 0.01 | 0.09 | | |
| Participant: Word Type (slope) | 0.001 | 0.03 | | |
| Participant: Exposure : Day (slope) | 0.01 | 0.12 | | |
| Participant: Day : Word Type (slope) | 0.01 | 0.07 | | |
| Item: (intercept) | 0.01 | 0.07 | | |
| Item: Day (slope) | 0.001 | 0.03 | | |

### 2.2.1. Semantic categorization (lexical integration)

Accuracy was close to ceiling across all conditions on Day 1 (standard exposure: $93.0 \pm 4.6\%$ (mean $\pm$ s.d.); high exposure: $92.0 \pm 5.3\%$) and Day 2 (standard exposure: $93.6 \pm 4.9\%$; high exposure: $92.3 \pm 5.2\%$). A mixed-effects linear model was fitted to the RT data (table 2). RTs were faster overall on Day 2 ($875 \pm 200$ ms) than Day 1 ($906 \pm 191$ ms) (figure 2), and responses were faster for hermits ($864 \pm 175$ ms) than for non-hermits ($918 \pm 212$ ms). Consistent with our hypotheses, Day and Word Type interacted such that the competition effect (i.e. the RT difference between non-hermits and hermits) was larger on Day 2 than on Day 1 (figure 2).

Also shown in table 2, there was a main effect of Delay, with shorter intervals between training and bedtime associated with faster RTs overall. However, there was no main effect of exposure, and removal of the Delay : Word Type : Day and Exposure : Word Type : Day fixed effects was justified ($p < 0.2$ for model comparisons). Thus, the change in lexical competition from Day 1 to Day 2 shown in figure 2 was not related to Exposure condition or Delay. However, there was a significant three-way Delay : Exposure : Day interaction (figure 3). As discussed, RTs on the whole tended to be about 30 ms faster

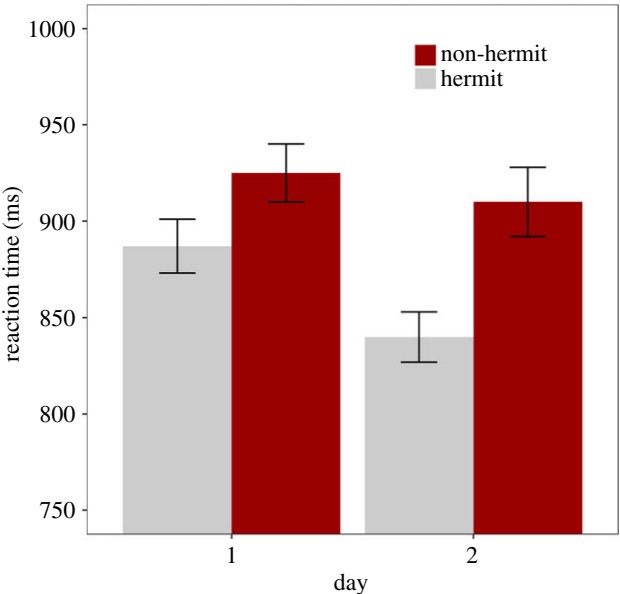

**Figure 2.** Semantic categorization mean reaction time (RT, in ms) for each Word Type (non-hermit, hermit) and Day (day 1, day 2). Error bars represent $\pm 1$ s.e. of the mean.

on Day 2 compared with Day 1. However, the three-way interaction shows that this effect was not uniform, with shorter delays leading to stronger RT improvements in the standard exposure condition, but longer delays having the same effect in the high exposure condition.

The Delay : Exposure : Day interaction was not predicted, and given that it does not involve the Word Type variable, it is of lesser importance for the current purposes. Nonetheless, the main effect of Delay (i.e. faster RTs for shorter delays) and particularly the interaction might have implications for our ability to test the influence of Exposure and Delay on the lexical competition effects on Day 2. Previous findings comparing adults and children suggest that faster RTs are associated with smaller competition effects [55], so it is important to ascertain whether such differences in RT might mask our ability to see an influence of Delay on changes in lexical competition across sleep. We tested whether participants' average RTs to the filler words in the current experiment were associated with their overall increase in lexical competition between Day 1 and Day 2. There was indeed a modest positive association, but this captured just 3% of the variation and was not statistically significant ($r = 0.174$, $p = 0.095$). Furthermore, the slope of this non-significant correlation (0.16) would suggest that the effects illustrated in figure 3 could only influence lexical competition by a few milliseconds. Therefore, we do not view these effects of Delay to be problematic in terms of impeding our main goal of determining the influence of Delay on consolidation-related changes in lexical competition.

### 2.2.2. Speeded recognition

Accuracy was high across all conditions on Day 1 ($96.2 \pm 6.8\%$) and Day 2 ($93.9 \pm 7.5\%$). A mixed-effects linear model was fitted to the speeded recognition RT data for correct responses only (table 3). The model revealed no significant effects.

### 2.2.3. Free recall

A mixed-effects generalized linear model (mixed logit) was fitted to the free recall accuracy data (table 4). As predicted, the model showed that accuracy was higher on Day 2 ($48.8 \pm 23.0\%$) than on Day 1 ($45.4 \pm 23.4\%$), but there were no effects of Exposure or Delay.

### 2.2.4. One-week re-test

Once testing was complete on Day 2 of Week 2, participants were re-tested on the words learned in Week 1. This was to determine whether any effects of encoding or delay on explicit memory of novel words, or on lexical integration, were present one week after training. It is important to note that there was no

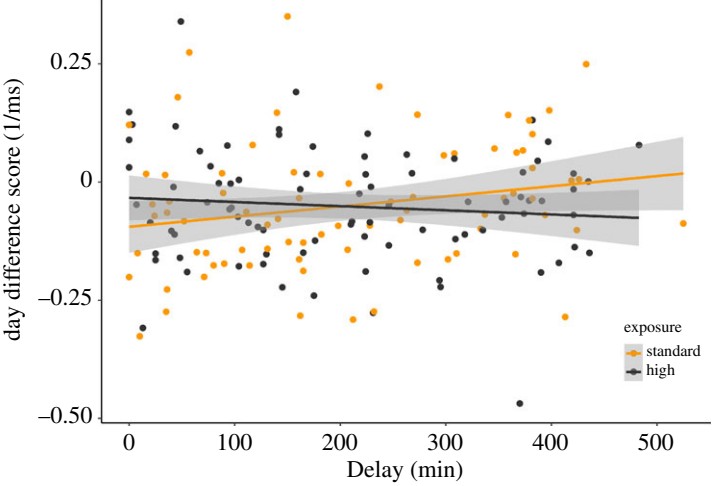

**Figure 3.** Scatterplot showing the correlation between Day Difference score (RTs on day 2 minus RTs on day 1, collapsed across Word Type condition) and Delay (in min) for each Exposure condition. Shaded areas represent 95% confidence intervals.

**Table 3.** Experiment 1 predictors of speeded recognition RT performance. Italics denote $p < 0.05$.

| fixed effects | $b$ | s.e. | $t$ | $p$ |
|---|---|---|---|---|
| *(Intercept)* | *−1.44* | *0.02* | *−72.21* | *<0.001* |
| Delay | 0.01 | 0.01 | 0.32 | 0.75 |
| Exposure | −0.01 | 0.02 | −0.71 | 0.48 |
| Day | 0.001 | 0.01 | 0.12 | 0.91 |
| Exposure : Day | 0.03 | 0.02 | 1.29 | 0.20 |
| random effects | variance | s.d. | | |
| Participant: (intercept) | 0.03 | 0.18 | | |
| Participant : Day (slope) | 0.01 | 0.09 | | |
| Participant : Exposure (slope) | 0.02 | 0.13 | | |
| Participant : Exposure : Day (slope) | 0.02 | 0.15 | | |
| Item : (intercept) | 0.002 | 0.05 | | |
| Item : Day (slope) | <0.001 | 0.03 | | |

one-week re-test for the words learned in Week 2. In order to retain power, the re-test data were not compared to the Week 1 data, as this would halve the number of re-test observations to be analysed.

A mixed-effects linear model was applied to the semantic categorization and speeded recognition RT re-test data (tables 5 and 6), and a mixed-effects generalized linear model applied to the free recall accuracy re-test data (table 7). Accuracy was high at follow-up for both the semantic categorization (90.3 ± 7.0) and speeded recognition tasks (91.4 ± 9.6), whereas it had decreased in the free recall task (25.8 ± 20.4). There were no main or interaction effects of Exposure evident across any of the tasks at re-test (tables 5–7).

For semantic categorization RT (table 5), there was a significant main effect of Word Type, such that responses were faster for hermits (940 ± 217 ms) than for non-hermits (1116 ± 324 ms). This suggests that evidence of lexical integration (i.e. the competition effect) was still present one week following training. There was no effect of Delay on semantic categorization RT performance at one-week re-test.

There was, however, a main effect of Delay on both speeded recognition RT performance (table 6) and free recall accuracy performance (table 7) at the one-week re-test. As presented in figures 4 and 5, there was a negative relationship between Delay and speeded recognition RT, and a positive relationship between Delay and free recall accuracy. Longer delays were associated with faster and more accurate responses. These findings provide preliminary suggestion that longer-term retention of explicit novel word memory might benefit from a longer period of wake following learning in adults.

**Table 4.** Experiment 1 predictors of free recall accuracy performance. Italics denote $p < 0.05$.

| fixed effects | $b$ | s.e. | $z$ | $p$ |
|---|---|---|---|---|
| (Intercept) | $-0.13$ | 0.13 | $-1.02$ | 0.31 |
| Delay | 0.04 | 0.10 | 0.41 | 0.68 |
| Exposure | $-0.02$ | 0.10 | $-0.20$ | 0.84 |
| *Day* | *0.18* | *0.05* | *3.33* | *<0.001* |
| Delay : Exposure | $-0.10$ | 0.11 | $-0.88$ | 0.38 |
| **random effects** | **variance** | **s.d.** | | |
| Participant : (intercept) | 1.15 | 1.07 | | |
| Participant : Exposure (slope) | 0.54 | 0.74 | | |
| Item : (intercept) | 0.24 | 0.49 | | |
| Item : Exposure1 : Delay (slope) | 0.03 | 0.17 | | |
| Item : Exposure2 : Delay (slope) | 0.08 | 0.28 | | |

**Table 5.** Experiment 1 predictors of semantic categorization RT performance at re-test. Italics denote $p < 0.05$.

| fixed effects | $b$ | s.e. | $t$ | $p$ |
|---|---|---|---|---|
| *(Intercept)* | *$-1.13$* | *0.03* | *$-42.86$* | *<0.001* |
| Exposure | 0.04 | 0.05 | 0.83 | 0.41 |
| Delay | 0.02 | 0.020 | 0.69 | 0.50 |
| *Word Type* | *1.12* | *0.02* | *6.61* | *<0.001* |
| Exposure : Word Type | $-0.03$ | 0.03 | $-0.89$ | 0.38 |
| **random effects** | **variance** | **s.d.** | | |
| Participant : (intercept) | 0.05 | 0.21 | | |
| Participant : Word Type (slope) | 0.01 | 0.09 | | |
| Item : (intercept) | 0.002 | 0.05 | | |
| Item : Word Type (slope) | 0.003 | 0.06 | | |

## 2.3. Discussion

In Experiment 1, we first aimed to replicate previous findings that the learning and integration of orthographic novel words benefits from a period of offline consolidation [19,21]. In addition, we explored whether these consolidation effects are influenced by the time between learning and sleep, and exposure level.

Importantly, our results replicated the finding that lexical competition between novel and existing words strengthens after a period of offline consolidation. This finding is in line with previous studies that emphasize the role of offline consolidation in enhancing the integration of novel words into the mental lexicon (e.g. [3,14,15,21,56]) as well as dual memory system theories of word learning (e.g. [16]). We also found that explicit word knowledge (as captured by a free recall task) improved overnight. By contrast, recognition speed did not change overnight; while this conflicts with previous reports of consolidation-related changes in speeded recognition of novel words [2], accuracy was very high and RT was relatively fast on this task with potentially little room for change.

Counter to previous claims [27–29], there was no evidence that less time awake between learning and sleep was beneficial for the overnight consolidation of novel orthographic forms in the present adult sample. Bolstering this conclusion, there was some tentative evidence that better recall and recognition of the novel words one week after training was associated with *more* time awake between training and sleep. Although requiring replication and further investigation, this finding possibly relates to claims made by Alger *et al.* [31], where a delay between learning and sleep can enhance declarative

**Table 6.** Experiment 1 predictors of speeded recognition RT performance at re-test. Italics denote $p < 0.05$.

| fixed effects | b | s.e. | t | p |
|---|---|---|---|---|
| (Intercept) | −0.14 | 0.03 | −49.20 | <0.001 |
| Exposure | −0.05 | 0.05 | −0.97 | 0.33 |
| Delay | −0.06 | 0.03 | −2.23 | 0.03 |
| random effects | variance | s.d. | | |
| Participant: (intercept) | 0.05 | 0.22 | | |
| Item: (intercept) | 0.003 | 0.26 | | |

**Table 7.** Experiment 1 predictors of recall accuracy performance at re-test. Italics denote $p < 0.05$.

| fixed effects | b | s.e. | T | p |
|---|---|---|---|---|
| (Intercept) | −1.44 | 0.20 | −7.19 | <0.001 |
| Exposure | 0.08 | 0.29 | 0.26 | 0.79 |
| Delay | 0.38 | 0.15 | 2.55 | 0.01 |
| random effects | variance | s.d. | | |
| Participant: (intercept) | 1.25 | 1.12 | | |
| Item: (intercept) | 0.70 | 0.84 | | |

memory retention. Alger *et al.* partly attributed this to increased opportunities for additional processing of the new stimuli prior to sleep. However, since the recall/recognition advantages for a longer delay only emerged one week later (and not on Day 2), it is likely that increased opportunities for wake-based processing in combination with experiencing the items again during the repeat recall/recognition tests on Day 2 led to the benefits one week later. Chiming with these results, Martini *et al.* [57] recently found that a post-encoding rest period led to enhanced memory 7 days later, but only when an additional recall took place at the end of the first experimental session (see also [58] for additive effects of intervening retrieval tests). Indeed, it has been argued that recall can trigger reconsolidation processes and facilitate neocortical integration [59].

Also counter to predictions, there were no effects of exposure level at training on any measure of word learning. Thus, it is plausible that both the standard and high exposure conditions were intensive enough to lead to robust representations that were too strong to be susceptible to wake-based interference or decay. Indeed, sleep consolidation effects have been shown to be reduced for stronger memory traces [27,36,37]. This is reinforced by participants' performance on the speeded recognition task, for which accuracy was at ceiling across all sessions. In Experiment 2 we therefore re-examined the hypotheses of Experiment 1 using a lower exposure level to test whether time awake has a stronger effect for more fragile initial memory traces.

# 3. Experiment 2

## 3.1. Method

### 3.1.1. Participants

One hundred and three monolingual English speakers with no reported reading, language, developmental or psychological disorders took part. Participants were students at the University of York aged 18–28 years (mean + 19.33 years, s.d. = 1.58 years; 12 males) and were paid or received course credits. There was no overlap in participation between Experiments 1 and 2.

Participants were asked to refrain from caffeine, alcohol and cigarettes on the day of training and the morning of testing. Twenty-five participants reported that they did not comply with these criteria and/or the task instructions, and were excluded from analysis.

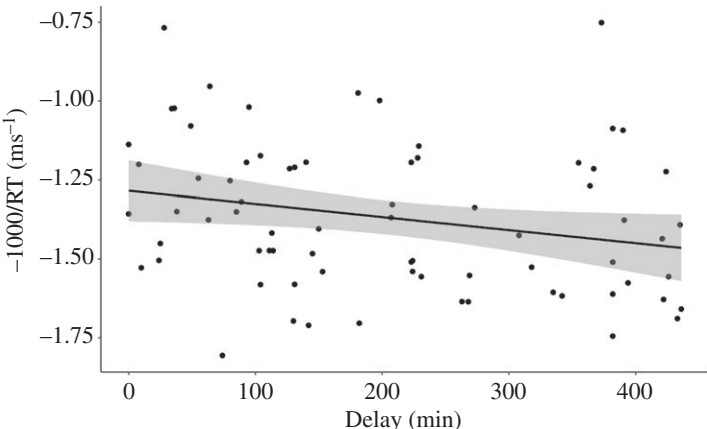

**Figure 4.** Speeded recognition mean inverse reaction time at re-test correlated with Delay collapsed across Exposure conditions. As Delay gets longer, RTs get faster.

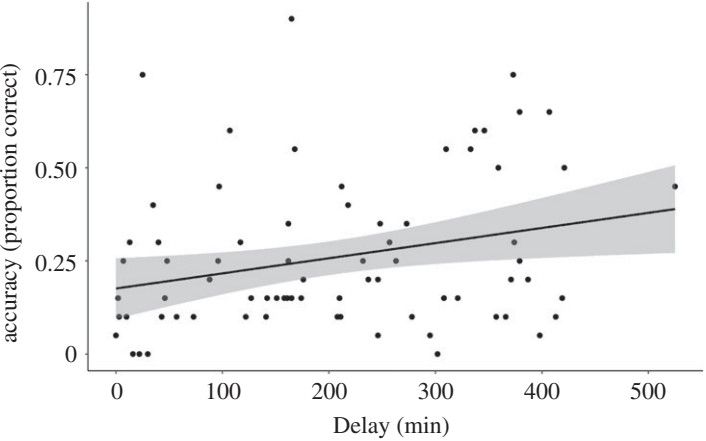

**Figure 5.** Recall mean per cent accuracy at re-test correlated with Delay collapsed across Exposure conditions.

### 3.1.2. Materials

The same stimuli and measures used in Experiment 1 were used in Experiment 2. Descriptives of the PSQI for Experiment 2 were similar to that of Experiment 1 (mean PSQI global score = 5.71, s.d. = 2.41, 1–14).

### 3.1.3. Procedure and design

The procedure and design for Experiment 2 was identical to Experiment 1, with the exception that there was only one 'Low' exposure condition, which had half as many training blocks as the Standard exposure condition from Experiment 1 (i.e. five blocks in total). The mean Delay was 204 min (s.d. = 147) and ranged from 0 to 527 min.

### 3.1.4. Analysis

Experiment 2 used the same method of analysis as Experiment 1.

*Outlier removal.* The same principles for data cleaning were applied as for Experiment 1. In the semantic categorization task, two participants were removed at the outlier removal stage (due to having accuracy rates 2.5 standard deviations below the mean), and six participants were removed at the Qualtrics check stage (due to Qualtrics server communication issues). One item was removed due to having accuracy rates 2.5 standard deviations below the mean (CHALK).

In the speeded recognition task, two participants were excluded due to response bias (identified as extreme low outliers in the d-prime distribution; d-prime cutoff 1.5), and 4/142 sessions of data had accuracy scores implausibly below chance (mean accuracy = 11.9%). As in Experiment 1, the response

**Table 8.** Experiment 2 mean (and s.d.) RTs for the semantic categorization and speeded recognition tasks (ms) and free recall accuracy (%). Italics denote $p < 0.05$.

| | semantic categorization | | | | |
| --- | --- | --- | --- | --- | --- |
| day | hermit | non-hermit | difference | speeded recognition | free recall |
| 1 | 894 (175) | 914 (191) | 20 (137) | 759 (102) | 32.9 (19.1) |
| 2 | 829 (169) | 851 (164) | 22 (113) | 735 (124) | 40.3 (21.6) |

key correspondence for these sessions was reversed. Six participants were removed due to Qualtrics server issues. In this task, three items were removed due to having accuracy rates 2.5 standard deviations below the mean (VELCET, BRETLE and BAMERY).

## 3.2. Results

Descriptive statistics for all main study variables can be found in table 8.

### 3.2.1. Semantic categorization (lexical integration)

Accuracy was high across all conditions on Day 1 ($92.4 \pm 7.1\%$) and Day 2 ($94.3 \pm 5.4\%$). A mixed-effects linear model was fitted to the semantic categorization RT data (table 9). As seen in figure 6, the model showed that RT was overall faster on Day 2 ($840 \pm 167$ ms) than Day 1 ($904 \pm 183$ ms) and faster to hermits ($861 \pm 175$ ms) than non-hermits ($882 \pm 180$ ms). The removal of the Word Type : Day fixed effect was justified (model comparison $p > 0.2$). As such, unlike that seen in Experiment 1, the competition effect did not increase overnight. Furthermore, Delay had no effect on the semantic categorization data, either as an interaction or a main effect.

### 3.2.2. Speeded recognition

Accuracy was again high across all conditions on Day 1 ($96.6 \pm 6.1\%$) and Day 2 ($94.1 \pm 6.4\%$). A mixed-effects linear model was fitted to the speeded recognition RT data (table 10). The model showed that RTs were faster on Day 2 in comparison to Day 1 (table 8 for descriptives).

### 3.2.3. Free recall

A mixed-effects generalized linear model (mixed logit) was fitted to the free recall accuracy data (table 11). The model showed that accuracy was higher on Day 2 in comparison to Day 1 (see table 8 for descriptives).

### 3.2.4. One-week re-test

Similar to Experiment 1, approximately one week after training, participants were re-tested on the words learned. Unlike Experiment 1, however, as there was only one Exposure condition in Experiment 2, all participants were trained on only one set of items, which were all re-tested the following week. Therefore we were able to modify the Day predictor to incorporate 3 levels (Day 1 versus Day 2 versus re-test) without having to reduce the number of observations. As recommended by Schad *et al.* [60], Day was coded using a variant of repeated contrasts, allowing us to compare Day 1 versus Day 2 (day 1 = $-1/3$, day 2 = $2/3$, re-test = $-1/3$), and Day 1 versus re-test: day 1 = $1/3$, day 2 = $1/3$, re-test = $-2/3$).

Accuracy was high at re-test for both the semantic categorization ($92.8 \pm 7.0\%$) and speeded recognition tasks ($90.5 \pm 9.8\%$), whereas it had slightly decreased in the free recall task ($31.9 \pm 46.6$).

As shown in table 12, there was no effect of Delay on semantic categorization RT performance at one-week re-test. There was, however, a significant main effect of Word Type, such that responses were faster for hermits ($862 \pm 141$ ms) than for non-hermits ($893 \pm 145$ ms). There was also a significant main effect of Day (1 versus 2), which was also seen in the main analysis (table 9).

As shown in tables 13 and 14, there were no predictive effects of Delay on participant's performance on the speeded recognition or free recall tasks at one-week re-test. The significant main effect of Day

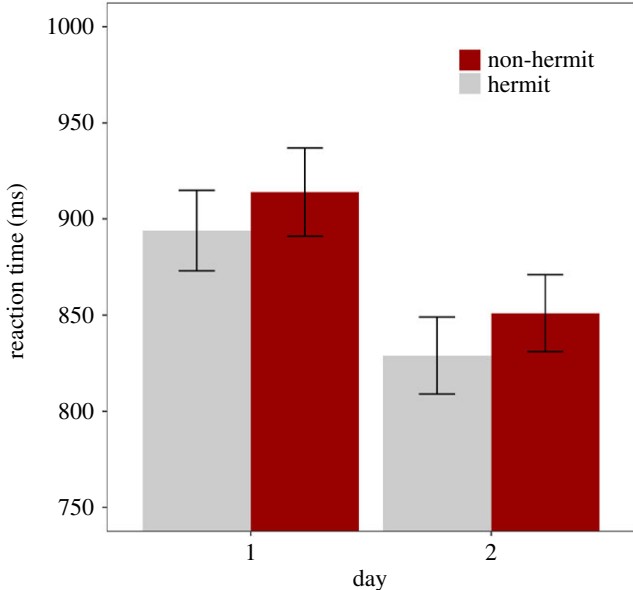

**Figure 6.** Semantic categorization mean reaction time (RT, in ms) for each Word Type (non-hermit, hermit) and Day (Day 1, Day 2). Error bars represent $\pm 1$ s.e. of the mean.

**Table 9.** Experiment 2 predictors of semantic categorization RT performance. Italics denote $p < 0.05$.

| fixed effects | b | s.e. | t | p |
|---|---|---|---|---|
| *(Intercept)* | *−1.25* | *0.02* | *−55.63* | *<0.001* |
| *Word Type* | *0.02* | *0.01* | *2.60* | *0.01* |
| Delay | 0.02 | 0.02 | 0.84 | 0.41 |
| *Day* | *−0.09* | *0.02* | *−5.81* | *<0.001* |
| random effects | variance | s.d. | | |
| Participant: (intercept) | 0.03 | 0.17 | | |
| Participant: Day (slope) | 0.01 | 0.11 | | |
| Participant: Word Type (slope) | 0.002 | 0.04 | | |
| Item: (intercept) | 0.01 | 0.07 | | |
| Item: Day (slope) | 0.001 | 0.03 | | |

(1 versus 2) was evident, as in the semantic categorization data, and is consistent with our earlier analysis (seen in tables 10 and 11).

## 3.3. Discussion

With a lower level of encoding in Experiment 2, an approximately 20 ms overall lexical competition effect was found following exposure to novel orthographic competitors, but this effect did not increase overnight. This compares with the 38 ms effect found after learning in Experiment 1, which increased to 82 ms the next day. Nevertheless, free recall accuracy and speeded recognition reaction times *did* improve after a period of sleep. Thus, while this level of encoding was sufficient for effects of offline consolidation to emerge for the explicit tasks (recall and recognition), it was not enough, apparently, to elicit any overnight change in lexical integration. Although previous studies of declarative memory have shown that weaker memory traces can benefit the most from sleep [27,37,38], it is possible that in our study the memory trace was *too* weak to benefit from sleep-associated consolidation, eliminating any overnight change in lexical integration effects. This is in line with research showing a lack of sleep-dependent lexical competition effects at low levels of exposure, even in the presence of

**Table 10.** Experiment 2 predictors of speeded recognition RT performance. Italics denote $p < 0.05$.

| fixed effects | $b$ | s.e. | $t$ | $p$ |
|---|---|---|---|---|
| *(Intercept)* | *−1.41* | *0.02* | *−70.14* | *<0.001* |
| Delay | 0.01 | 0.02 | 0.45 | 0.66 |
| *Day* | *−0.05* | *0.02* | *−2.73* | *0.01* |
| Day : Delay | −0.03 | 0.02 | −1.59 | 0.12 |
| random effects | variance | s.d. | | |
| Participant: (intercept) | 0.02 | 0.15 | | |
| Participant: Day (slope) | 0.02 | 0.13 | | |
| Item: (intercept) | 0.01 | 0.07 | | |
| Item: Delay (slope) | <0.001 | 0.02 | | |

**Table 11.** Experiment 2 predictors of free recall accuracy performance. Italics denote $p < 0.05$.

| fixed effects | $b$ | s.e. | $z$ | $p$ |
|---|---|---|---|---|
| *(Intercept)* | *−0.72* | *0.14* | *−5.03* | *<0.001* |
| Delay | 0.03 | 0.13 | 0.21 | 0.83 |
| *Day* | *0.41* | *0.09* | *4.90* | *<0.001* |
| random effects | variance | s.d. | | |
| Participant: (intercept) | 0.94 | 0.97 | | |
| Item: (intercept) | 0.50 | 0.70 | | |
| Item: Delay (slope) | 0.13 | 0.36 | | |

good recognition performance [14,61]. It is also consistent with a recent study of children with dyslexia who showed poorer levels of encoding and no overnight emergence of lexical competition in response to novel spoken competitors, despite showing overnight improvements in explicit recall [7]. Taken together, these studies suggest that variations in encoding conditions are an important determiner of overnight consolidation effects, with lexical integration being particularly susceptible to these variations.

Importantly, with less intensive encoding procedures, the time between learning and sleep did not influence overnight consolidation. Moreover, unlike Experiment 1, there was no suggestion of a benefit of increased opportunities for wake-based processing prior to sleep on recall or recognition at the one-week follow-up. This could be a consequence of lower overall levels of explicit novel word memory in Experiment 2 relative to Experiment 1, which may have reduced opportunities for explicit wake-based processing and/or reconsolidation following the Day 2 re-test. In accordance with this, it is well established that less stable memory representations are more vulnerable to wake-based interference immediately after their acquisition [62,63].

# 4. General discussion

This study set out to examine variables that influence the integration of novel orthographic forms, as well as more standard tests of explicit memory for these forms. Adopting a training procedure known to show sleep-related benefits in the lexical integration of pseudowords (e.g. BANARA; Bowers *et al.* [19]; Wang *et al.* [21]), we manipulated both exposure level and the time between learning and sleep. Experiment 1 showed that overnight consolidation of novel orthographic forms (e.g. BANARA) leads to influences on the processing of existing form neighbours (e.g. BANANA) that are enhanced on the day after learning. This is largely in line with previous findings [19,21] and lends further support to claims that the CLS approach to vocabulary learning [16] applies to written as well as spoken word learning. Under this premise, we suggest that the overnight enhancement of competition effects may be due to the sleep-dependent benefits of offline consolidation, allowing for the integration of the

**Table 12.** Experiment 2 predictors of semantic categorization RT performance at re-test. Italics denote $p < 0.05$.

| fixed effects | b | s.e. | t | p |
|---|---|---|---|---|
| (*Intercept*) | *−1.24* | *0.02* | *−59.46* | *<0.001* |
| Delay | 0.02 | 0.02 | 0.88 | 0.38 |
| *Word Type* | *0.03* | *0.01* | *4.54* | *<0.001* |
| *Day (1 versus 2)* | *−0.09* | *0.02* | *−5.82* | *<0.001* |
| Day (1 versus re-test) | 0.03 | 0.02 | 1.69 | 0.10 |
| Delay : Day (1 versus 2) | −0.004 | 0.02 | −0.24 | 0.81 |
| Delay : Day (1 versus re-test) | −0.01 | 0.02 | −0.69 | 0.49 |
| **random effects** | **variance** | **s.d.** | | |
| Participant: (intercept) | 0.03 | 0.16 | | |
| Participant : Day (1 versus 2) (slope) | 0.01 | 0.11 | | |
| Participant : Day (1 versus re-test) (slope) | 0.02 | 0.12 | | |
| Item: (intercept) | 0.004 | 0.07 | | |
| Item : Day (1 versus 2) (slope) | <0.001 | 0.03 | | |
| Item : Day (1 versus re-test) (slope) | <0.001 | 0.03 | | |

**Table 13.** Experiment 2 predictors of speeded recognition RT performance at re-test. Italics denote $p < 0.05$.

| fixed effects | b | s.e. | t | p |
|---|---|---|---|---|
| (*Intercept*) | *−1.40* | *0.02* | *−72.63* | *<001* |
| *Day (1 versus 2)* | *−0.05* | *0.02* | *−2.66* | *0.01* |
| Day (1 versus re-test) | −0.001 | 0.02 | 0.03 | 0.98 |
| Delay | 0.002 | 0.02 | 0.12 | 0.91 |
| Delay : Day (1 versus 2) | −0.02 | 0.02 | 1.08 | 0.29 |
| Delay : Day (1 versus re-test) | 0.03 | 0.02 | 1.46 | 0.15 |
| **random effects** | **variance** | **s.d.** | | |
| Participant: (intercept) | 0.02 | 0.14 | | |
| Participant : Day (1 versus 2) (slope) | 0.02 | 0.13 | | |
| Participant : Day (1 versus re-test) (slope) | 0.02 | 0.15 | | |
| Item: (intercept) | 0.01 | 0.07 | | |

novel word into the lexicon [3,14,15]. Moreover, we can extend these findings to highlight that such effects appear to be limited by levels of exposure. Specifically, when exposure was twice the typical level used in experiments of this type (20 rather than 10 exposures to each novel word), there was no significant alteration of the overnight boost to the test of lexical integration (Experiment 1). However, when the encoding opportunity was more limited (five exposures to each novel word), the overnight strengthening of lexical integration was not observed (Experiment 2). This has potential implications for individuals with language learning difficulties, for whom lexical integration may be compromised during sleep, at least partly as a consequence of encoding differences [7].

Intriguingly, the lower exposure limit on consolidation effects for our measure of lexical engagement was not seen for the more explicit tests of memory for the novel words. That is, overnight improvements in adults' ability to recall the novel words were observed at all levels of exposure. There are several possible explanations for this dissociation. It could be that lexical integration has a higher initial encoding threshold than simple memory for novel items, or that the explicit tests are more sensitive. It could also be related to the fact that the tests were repeated with the same items on both days, leading to retrieval practice effects for the explicit tests (e.g. [64]). Retrieval practice is likely to be of

**Table 14.** Experiment 2 predictors of free recall accuracy performance at re-test. Italics denote $p < 0.05$.

| fixed effects | b | s.e. | z | p |
|---|---|---|---|---|
| (Intercept) | −0.83 | 0.15 | −5.47 | <0.001 |
| Day (1 versus 2) | 0.44 | 0.09 | 5.05 | <0.001 |
| Day (1 versus re-test) | 0.06 | 0.09 | 0.63 | 0.53 |
| Delay | 0.01 | 0.14 | 0.05 | 0.96 |
| random effects | variance | s.d. | | |
| Participant: (intercept) | 1.09 | 1.05 | | |
| Item: (intercept) | 0.54 | 0.74 | | |
| Item : Delay (slope) | 0.16 | 0.40 | | |

less benefit for the semantic categorization task that underlies the test of lexical integration, given that the item presented is not the item that has recently been learned and the effect depends on a decrement in performance (i.e. slower RTs to the test items) rather than an improvement.

In contrast to our initial predictions and previous claims, there were no clear effects of the delay between learning and bedtime on overnight consolidation. It is possible that pre-sleep–wake effects in the present and previous studies may be contaminated by time of day effects. Although in the current study individual training sessions did occur at different times in the day as Delay was dependent on participants' own bedtimes, exploration of the vigilance task data showed no impact of attention on the relationship between Delay and the other dependent variables. Moreover, previous research has shown that circadian rhythms do not explain differences in test/re-test performance [3,65]. Therefore, it is unlikely that time of day effects can explain the present findings.

Despite not seeing clear effects of time awake prior to sleep on Day 2 performance in the present study, we did observe a significant predictive role of Delay at the one-week test, with more time awake between learning and sleep being associated with better recall and recognition one week later. As discussed briefly above, this resonates with Alger et al.'s [31] suggestion that increased time awake leads to increased opportunities for wake-based processing prior to sleep (see also [66]). However, our data do not directly align with this claim, since we only observed a benefit at one week and not on Day 2 and we did not observe the same benefit under lower exposure conditions in Experiment 2. Thus, it seems plausible that a combination of wake-based processing *and* the repeat recall/ recognition testing on Day 2 led to enhanced explicit memory one week later (see also [57]), at least when explicit memory reached a level sufficient to engage such processing. However, further research is needed to examine this hypothesis specifically.

There are a number of contexts in which it will be important to further explore the effects of the time between learning and sleep. For instance, it is possible that effects of time awake could be sensitive to individual differences in prior knowledge. That is, time awake before sleep may be less important in adults for language learning as they have a strong existing knowledge base to support learning and/ or consolidation which may overcome effects of wake-based interference [9]. Thus, an exploration of the effects of time awake in adult studies where prior knowledge cannot support task performance to the same degree would be fruitful (e.g. in studies of word learning that manipulate prior knowledge, or in studies of spatial declarative memory or sequence learning; see James et al. [9] for a discussion). Further to this issue, waking time effects may also be developmentally sensitive. In contrast to adults, it may be that children are more susceptible to wake-based interference as a consequence of lower levels of existing knowledge and/or protracted hippocampal development [9,67,68]. Research investigating child versus adult differences thus presents a further important avenue.

We also acknowledge that the less tightly controlled nature of the online testing environment, while advantageous in many ways, may have meant that participants did not adhere to the task instructions as well as would have been expected in the laboratory. Certainly, the rate of data loss in the current study was higher than the amount of data loss we would normally see in our previous lab-based studies. Nevertheless, online testing permitted a naturalistic exploration of time awake right up until participants' typical bedtimes, and was therefore necessary for the current purposes. It is vital, however, that with online testing becoming increasingly popular, experimenters take appropriate precautions and comply with the appropriate data checks once testing is complete (see [43]).

It is also important for us to address the limitation of both experiments in not including a wake control group. While we can reasonably suppose that the overnight change in lexical competition is due to consolidation during sleep, it is also possible that wake consolidation could be influential [69]. The closest equivalent to a wake control in the literature is the study of Wang et al. [21]. Their study used materials and methods very similar to our own, and found that a period of 12 h awake across the day had no impact on the strength of lexical competition for visually presented words. This evidence supports our interpretation of our own results as relating to consolidation in sleep, but it remains possible that implementational differences between the two studies could have been influential (for example their study was lab-based rather than web-based).

Although a clear overnight increase in lexical competition was evident for Experiment 1, there were also small competition effects observed on Day 1 in Experiments 1 and 2. This potentially conflicts with the predictions of the CLS framework (in which lexical integration should require a period of offline consolidation to emerge), and with previous findings [3,19,21]. One possibility is that these weak Day 1 effects could arise from the generally slower RTs that were obtained in this study, perhaps as a consequence of the online testing procedure. For example, mean RTs on Day 1 for Hermits were 864 ms (Experiment 1) and 894 ms (Experiment 2) and for Non-hermits, 918 ms (Experiment 1) and 914 ms (Experiment 2); whereas in Bowers et al. [19], who tested participants in the lab, mean RTs on Day 1 were 730 ms and 747 ms to hermits and non-hermits, respectively. Previous studies have shown that slower RTs are associated with numerically larger lexical competition effects [55]. This could be simply because there is greater room for a difference to emerge with slower RTs, and/or because processing may become more strategic, allowing participants to consciously note the overlap between novel and existing words. However, there were no correlations between the size of the lexical competition effects on Day 1 and participants' mean RTs to the filler items (Experiment 1: $r = 0.05$, $p = 0.66$; Experiment 2: $r = 0.12$, $p = 0.31$). Thus, there was no evidence that slower RTs were accounting for the weak Day 1 lexical competition effects here. While there was a correlation between lexical competition on Day 2 and participants' mean filler RT for Experiment 1 ($r = 0.34$, $p < 0.001$), this correlation was not significant for Experiment 2 ($r = -0.07$, $p = 0.56$). Therefore, only where we observed an overnight emergence of lexical competition (i.e. Day 2, Experiment 1) did participants' with slower RTs show larger effects. It is also important to note that slower RTs cannot solely account for the overnight emergence of lexical competition observed in Experiment 1, since participants' RTs were faster on Day 2 than on Day 1. Thus, while slower RTs influenced the size of the competition effect, they cannot account for the emergence of the effect, which we attribute to the process of lexical integration.

Instead it may be that both visual and spoken word recognition tests of engagement in lexical competition should not be thought of as all or nothing, as previous results might suggest [14]. In fact, in the original visual study [19], there was a numerical competition effect of 17 ms at the first test point soon after exposure. This was not statistically significant, but is similar in size to the significant 20 ms effect that we found in Experiment 2. Therefore, the key test of consolidation-related lexical integration may not be the presence or the absence of any competition effect, but the size of this effect, and whether it is enhanced by sleep [21] or is associated with sleep parameters [2,7]. This is in fact consistent with more detailed accounts of how a CLS model of word learning might operate [16]. As discussed in McMurray et al. [70], the difference between pre-consolidation and post-consolidation representations of novel words is not one of encapsulation of the new word versus integration with its lexical neighbours; rather, it is a difference in efficiency of mapping between form and meaning [16], with hippocampal mediation seen as less efficient, slower and less automatic [71,72].

The latter characterization of novel word competition effects has ramifications for studies that have used the effect as a measure of immediate lexical integration. Particularly, Coutanche & Thompson-Schill [10] have used the Bowers et al. [19] methodology as a means of assessing the consequences of different types of learning. They assessed a 'fast mapping' procedure [73] that required participants to infer the referent of a novel word such as BANARA in the context of a visual display containing a familiar object and a novel object. In these circumstances, participants should associate the novel word with the novel object, and this type of association has been argued to be less dependent on the hippocampus than explicit encoding [73]. Coutanche & Thompson-Schill found that fast mapping but not explicit encoding led to a visual word competition effect on the same day of testing, which they took as evidence of lexical integration. While this dissociation is undoubtedly interesting and potentially important, the inference that fast mapping has led to immediate lexical integration is, in the light of the above discussion, weakened (see also [74–76] for further discussion of this issue).

In conclusion, this study replicates previous findings that the learning and integration of novel orthographic forms benefits from offline consolidation. We demonstrate that the integration of novel

words can be compromised by low exposure conditions, and importantly, once a novel word is encountered sufficiently to initiate offline integration, there is no further benefit of higher levels of exposure. Counter to commonly cited claims, there was no evidence that less time awake following learning was beneficial to the consolidation of novel words in the present adult sample; indeed, there was some evidence that *more* time awake following learning can benefit explicit aspects of novel word memory one week after training (at least under higher exposures conditions). Such findings emphasize the need to further explore the variables that lead to individual differences in the time course of lexical integration, to advance our theoretical understanding of novel word learning and ultimately inform practice.

Ethics. All experiments were approved by the University of York Psychology Ethics Committee.

Data accessibility. All data are available at: https://osf.io/w2u7j/. The Experimental Stimuli have been uploaded as part of the electronic supplementary material.

Authors' contributions. S.W. contributed to the design of the study, carried out the data acquisition, analysis and interpretation of the data and drafted the manuscript; L.H. contributed to the conception, design, analysis and interpretation of the data and helped draft the manuscript; F.F. contributed to the design, analysis and interpretation of the data and helped draft the manuscript; V.K. contributed to the design of the study, the interpretation of the data, and helped draft the manuscript; S.C. contributed to the conception and design of the study, and helped draft the manuscript; M.G. contributed to the conception and design of the study, the analysis and interpretation of the data, and helped draft the manuscript. All authors gave final approval for publication.

Competing interests. We declare we have no competing interests.

Funding. The research was supported by UK Economic and Social Research Council grant no. ES/N009924/1 (awarded to Lisa Henderson, Gareth Gaskell and Courtenay Norbury).

Acknowledgements. First, we thank the participants for taking part in this research. We are very grateful to members of the Sleep, Language and Memory Lab for valuable discussion of this work.

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
