## [Reviewer comments · Royal Society Open Science]

Review History

RSOS-181842.R0 (Original submission)

Review form: Reviewer 1 (Anne Castles)

Is the manuscript scientifically sound in its present form?

Yes

Are the interpretations and conclusions justified by the results?

Yes

Is the language acceptable?

Yes

Is it clear how to access all supporting data?

Yes

Do you have any ethical concerns with this paper?

No

Have you any concerns about statistical analyses in this paper?

Yes

Recommendation?

Major revision is needed (please make suggestions in comments)

Comments to the Author(s)

This paper reports on two experiments exploring factors affecting the consolidation of new vocabulary, and specifically the effects of exposure level and delay between learning and sleep. These are important questions and will be of interest to readers of RSOS. The experimental design and analyses seem generally sound (although I have one major comment, outlined below), and the paper is well written. The pattern of results across the two experiments is rather complicated, but the authors do a good job of interpreting them and placing them in the broader context of other research in the field.

I really have only one major concern that I think needs to be addressed, and that relates to the removal of quite a large amount of data due to non-compliance of participants with the instructions to avoid caffeine, alcohol and cigarettes. In the first experiment, the data of 20 participants were removed across Weeks 1 and 2, and in the second experiment 25/103 participants were excluded. I question this decision for a number of reasons. First of all, I wondered why such an exclusion might have been applied in the first place? If the participants were habitual smokers or caffeine or alcohol drinkers, might asking them to abstain not have interrupted their usual nocturnal sleep processes? I would have thought it would be better to have participants engage in their typical activities, and modify their behaviour as little as possible (obviously avoiding extreme behaviour such as binge drinking etc). I could not find any reference to a justification for these strict exclusion criteria, and it did not appear to be mentioned in the preregistration document for the first experiment.

Secondly, I would be concerned that the participants who had data removed may have differed from the included participants in other systematic ways that might be relevant to the present findings. For example, they may have been participants who were less committed to the experiment, and to the training required, and who therefore may have formed weaker lexical traces. Given that strength of initial encoding is one of the factors of interest in this study, it would seem important that this kind of individual variation within the population is captured.

At the very least, I think that a much stronger justification needs to be provided of the exclusion criteria, weighing the removal of so much data up against the significant loss of power and the potential for the remaining data to be non-representative. I also think the data should be analysed with and without the excluded data. If there are no differences in the overall pattern of results, then the full set should be reported on (with a footnote explaining this). If there are differences, then both sets of analyses should be reported on, and the possible bases for any differences in results discussed.

Minor points:

- “Complementary Learning Systems” is sometimes capitalised and sometimes not.
- p.6, line 12: “stronger acquisition of a motor adaption task”. I didn’t know what this meant.
- p. 13, line 5: “Participants were presented with the newly learned items mixed with foils”. What were the foils here, and how constructed?

- Table 1 and 2 (and throughout): There are a very large number of tables, but many only include a few numbers and could easily be merged with other tables.

- p. 21, line 40: Was Experiment 2 preregistered? Why Experiment 1 but not 2?

Review form: Reviewer 2

Is the manuscript scientifically sound in its present form?

Yes

Are the interpretations and conclusions justified by the results?

Yes

Is the language acceptable?

Yes

Is it clear how to access all supporting data?

No

Do you have any ethical concerns with this paper?

No

Have you any concerns about statistical analyses in this paper?

No

Recommendation?

Accept with minor revision (please list in comments)

Comments to the Author(s)

This study investigates the conditions under which novel competitors to existing words induce lexical competition across a period including sleep. Two factors are manipulated: the time between learning and sleep, and the amount of exposure to the novel words. Previously observed increases in lexical competition across sleep were replicated for standard and high exposure levels, but not for lower exposure levels. For the higher exposure levels, longer gaps between learning and sleep were associated with better explicit memory of novel words one week later.

A strength of this paper is its pre-registration, which makes transparent the predictions and experimental plans. The analysis strategy taken in the paper deviated from the pre-registration, however, which raises the question of whether the authors additionally ran the originally-planned analyses (if so, they should be reported).

Perhaps the biggest weakness of the paper is the lack of wake control groups, which limits the ability to make claims about the impact of sleep (as opposed to time passing). Though it would be a stronger paper with those groups, it is a solid enough contribution without them. This limitation deserves more explicit discussion, however.

The unpredicted relationship between delay and memory one week later with higher exposure levels appears to be weak and its mechanism is unclear; it seems likely to be spurious. I would encourage the authors to treat these results with even more caution than they are currently

treated (for example, in the abstract, perhaps stating that “there was no clear relationship” between delay and memory change, as opposed to taking seriously the complicated pattern of results).

It was very difficult to flip back and forth to the tables, figures, and figure legends; embedded figures and tables would facilitate review.

It is not clear that the analysis code is / will be made available.

Review form: Reviewer 3

Is the manuscript scientifically sound in its present form?

No

Are the interpretations and conclusions justified by the results?

Yes

Is the language acceptable?

Yes

Is it clear how to access all supporting data?

Yes

Do you have any ethical concerns with this paper?

No

Have you any concerns about statistical analyses in this paper?

Yes

Recommendation?

Major revision is needed (please make suggestions in comments)

Comments to the Author(s)

This manuscript examines how level of exposure to novel words, and the delay between learning and sleep, influence subsequent sleep-dependent consolidation of the novel words to existing words. These are key issues within the sleep and memory/learning literature, and the findings of these experiments are important and would be of interest to the readers of the journal. However, I do have some concerns with the manuscript which I believe can be addressed in a revision.

General comments

The study uses a really nice sample size, but what was the motivation for testing 113 participants? The manuscript states the preregistered target was 80, it might be worth mentioning a bit more about where that target came from, and make it clearer why data collection wasn't stopped when this number was reached.

It's not entirely clear that the study was completed online via qualtrics until around page 10. This should be clarified, and perhaps raised earlier on as one of the contributions of this literature to the field - many of the prior studies of sleep/lexical integration have involved laboratory-style tests - the value of implementing this online should be highlighted.

Some more detailed motivation for including the semantic categorisation task is needed (p7 par 1), and perhaps earlier on too. How/why is it a marker of lexical integration? You could discuss this in more detail on p4 par 2. This paragraph needs some restructuring to make it clearer how the semantic categorisation task is a marker of lexical integration.

The reversal of responses on the speeded recognition task is a bit concerning (while I'm sympathetic to the logic). It might be safer to avoid this. My recommendation is to exclude these participants' data from analysis if they are deemed to be outliers, since we can only be so confident in this assumption. Given the great sample size here, there should be sufficient power to do this and still have a good amount of data left over.

I am not familiar with using $p < .2$ as a threshold for justification of fixed/random effects in a model. Can you give readers more detail about how you came to this decision?

Page 10 - 11; how certain can we be about participants' adherence to these instructions - I imagine you checked the test time for each participant to make sure they stuck to the schedule? This kind of detail should be reported. Similarly, did you collect information on self-reported sleep duration in between day 1 and day 2? Was this similar across the different groups?

Minor comments

Learning to live with interfering neighbours: The influence of time of learning and level of Encoding.. Title - the last part feels incomplete? On what - lexical integration?

Page 2 line 13: "Participants made speeded semantic decisions (i.e., natural/artefact) to the existing words..." - ? this is unclear

Page 4 line 28: "Bowers et al. trained novel word forms..." - incomplete sentence? This section needs reworking.

Page 4 line 38: response times were slower to the words... - slower for what? Was this for accurate responses only?

Page 7 line 19: clarify which kind of online you mean here

Page 7 line 33: a more intensive level... of exposure? Incomplete sentence

Page 14 line 47: there is a rogue bracket

Decision letter (RSOS-181842.R0)

11-Feb-2019

Dear Dr Walker,

The editors assigned to your paper ("Learning to live with interfering neighbours: The influence of time of learning and level of encoding") have now received comments from reviewers. We would like you to revise your paper in accordance with the referee and Associate Editor suggestions which can be found below (not including confidential reports to the Editor). Please note this decision does not guarantee eventual acceptance.

Please submit a copy of your revised paper before 06-Mar-2019. Please note that the revision deadline will expire at 00.00am on this date. If we do not hear from you within this time then it will be assumed that the paper has been withdrawn. In exceptional circumstances, extensions may be possible if agreed with the Editorial Office in advance. We do not allow multiple rounds of revision so we urge you to make every effort to fully address all of the comments at this stage.

If deemed necessary by the Editors, your manuscript will be sent back to one or more of the original reviewers for assessment. If the original reviewers are not available, we may invite new reviewers.

- Data accessibility

If you wish to submit your supporting data or code to Dryad (<http://datadryad.org/>), or modify your current submission to dryad, please use the following link:
<http://datadryad.org/submit?journalID=RSOS&manu=RSOS-181842>

- Competing interests

- Authors' contributions

- Acknowledgements

- Funding statement

Kind regards,

Andrew Dunn

on behalf of Professor Carolyn McGettigan (Associate Editor) and Essi Viding (Subject Editor)

Associate Editor's comments (Professor Carolyn McGettigan):

Associate Editor: 1

Comments to the Author:

Once again please accept our apologies at the length of the review process on this occasion - it took some time to identify suitable reviewers in the first instance but I am pleased that we have now received comments on your manuscript from 3 relevant experts in the field.

You will see that all 3 reviewers were pleased with the manuscript, finding it of interest and of relevance to the readership of the journal. They each also expressed some concerns, which although not huge in number do include a request for re-analysis (see Reviewer 1) as well as some additional discussion of some of the design and analysis choices. Given that the impact of re-analysis on the interpretation of the findings is currently unknown, I am technically requesting a major revision at this point as a "worst case scenario" - the authors may find that the issues can be addressed more straightforwardly than this official decision suggests.

Comments to Author:

Reviewers' Comments to Author:

Reviewer: 1

Comments to the Author(s)

This paper reports on two experiments exploring factors affecting the consolidation of new vocabulary, and specifically the effects of exposure level and delay between learning and sleep. These are important questions and will be of interest to readers of RSOS. The experimental design

and analyses seem generally sound (although I have one major comment, outlined below), and the paper is well written. The pattern of results across the two experiments is rather complicated, but the authors do a good job of interpreting them and placing them in the broader context of other research in the field.

I really have only one major concern that I think needs to be addressed, and that relates to the removal of quite a large amount of data due to non-compliance of participants with the instructions to avoid caffeine, alcohol and cigarettes. In the first experiment, the data of 20 participants were removed across Weeks 1 and 2, and in the second experiment 25/103 participants were excluded. I question this decision for a number of reasons. First of all, I wondered why such an exclusion might have been applied in the first place? If the participants were habitual smokers or caffeine or alcohol drinkers, might asking them to abstain not have interrupted their usual nocturnal sleep processes? I would have thought it would be better to have participants engage in their typical activities, and modify their behaviour as little as possible (obviously avoiding extreme behaviour such as binge drinking etc). I could not find any reference to a justification for these strict exclusion criteria, and it did not appear to be mentioned in the preregistration document for the first experiment.

Secondly, I would be concerned that the participants who had data removed may have differed from the included participants in other systematic ways that might be relevant to the present findings. For example, they may have been participants who were less committed to the experiment, and to the training required, and who therefore may have formed weaker lexical traces. Given that strength of initial encoding is one of the factors of interest in this study, it would seem important that this kind of individual variation within the population is captured.

At the very least, I think that a much stronger justification needs to be provided of the exclusion criteria, weighing the removal of so much data up against the significant loss of power and the potential for the remaining data to be non-representative. I also think the data should be analysed with and without the excluded data. If there are no differences in the overall pattern of results, then the full set should be reported on (with a footnote explaining this). If there are differences, then both sets of analyses should be reported on, and the possible bases for any differences in results discussed.

Minor points:

- “Complementary Learning Systems” is sometimes capitalised and sometimes not.
- p.6, line 12: “stronger acquisition of a motor adaption task”. I didn’t know what this meant.
- p. 13, line 5: “Participants were presented with the newly learned items mixed with foils”. What were the foils here, and how constructed?
- Table 1 and 2 (and throughout): There are a very large number of tables, but many only include a few numbers and could easily be merged with other tables.
- p. 21, line 40: Was Experiment 2 preregistered? Why Experiment 1 but not 2?

Reviewer: 2

Comments to the Author(s)

This study investigates the conditions under which novel competitors to existing words induce lexical competition across a period including sleep. Two factors are manipulated: the time

between learning and sleep, and the amount of exposure to the novel words. Previously observed increases in lexical competition across sleep were replicated for standard and high exposure levels, but not for lower exposure levels. For the higher exposure levels, longer gaps between learning and sleep were associated with better explicit memory of novel words one week later.

A strength of this paper is its pre-registration, which makes transparent the predictions and experimental plans. The analysis strategy taken in the paper deviated from the pre-registration, however, which raises the question of whether the authors additionally ran the originally-planned analyses (if so, they should be reported).

Perhaps the biggest weakness of the paper is the lack of wake control groups, which limits the ability to make claims about the impact of sleep (as opposed to time passing). Though it would be a stronger paper with those groups, it is a solid enough contribution without them. This limitation deserves more explicit discussion, however.

The unpredicted relationship between delay and memory one week later with higher exposure levels appears to be weak and its mechanism is unclear; it seems likely to be spurious. I would encourage the authors to treat these results with even more caution than they are currently treated (for example, in the abstract, perhaps stating that “there was no clear relationship” between delay and memory change, as opposed to taking seriously the complicated pattern of results).

It was very difficult to flip back and forth to the tables, figures, and figure legends; embedded figures and tables would facilitate review.

It is not clear that the analysis code is / will be made available.

Reviewer: 3

Comments to the Author(s)

This manuscript examines how level of exposure to novel words, and the delay between learning and sleep, influence subsequent sleep-dependent consolidation of the novel words to existing words. These are key issues within the sleep and memory/learning literature, and the findings of these experiments are important and would be of interest to the readers of the journal. However, I do have some concerns with the manuscript which I believe can be addressed in a revision.

General comments

The study uses a really nice sample size, but what was the motivation for testing 113 participants? The manuscript states the preregistered target was 80, it might be worth mentioning a bit more about where that target came from, and make it clearer why data collection wasn't stopped when this number was reached.

It's not entirely clear that the study was completed online via qualtrics until around page 10. This should be clarified, and perhaps raised earlier on as one of the contributions of this literature to the field – many of the prior studies of sleep/lexical integration have involved laboratory-style tests – the value of implementing this online should be highlighted.

Some more detailed motivation for including the semantic categorisation task is needed (p7 par 1), and perhaps earlier on too. How/why is it a marker of lexical integration? You could discuss this in more detail on p4 par 2. This paragraph needs some restructuring to make it clearer how the semantic categorisation task is a marker of lexical integration.

The reversal of responses on the speeded recognition task is a bit concerning (while I'm sympathetic to the logic). It might be safer to avoid this. My recommendation is to exclude these participants' data from analysis if they are deemed to be outliers, since we can only be so confident in this assumption. Given the great sample size here, there should be sufficient power to do this and still have a good amount of data left over.

I am not familiar with using $p < .2$ as a threshold for justification of fixed/random effects in a model. Can you give readers more detail about how you came to this decision?

Page 10 – 11; how certain can we be about participants' adherence to these instructions – I imagine you checked the test time for each participant to make sure they stuck to the schedule? This kind of detail should be reported. Similarly, did you collect information on self-reported sleep duration in between day 1 and day 2? Was this similar across the different groups?

Minor comments

Learning to live with interfering neighbours: The influence of time of learning and level of Encoding.. Title – the last part feels incomplete? On what - lexical integration?

Page 2 line 13: "Participants made speeded semantic decisions (i.e., natural/artefact) to the existing words...." – ? this is unclear

Page 4 line 28: "Bowers et al. trained novel word forms..." – incomplete sentence? This section needs reworking.

Page 4 line 38: response times were slower to the words... – slower for what? Was this for accurate responses only?

Page 7 line 19: clarify which kind of online you mean here

Page 7 line 33: a more intensive level... of exposure? Incomplete sentence

Page 14 line 47: there is a rogue bracket

Author's Response to Decision Letter for (RSOS-181842.R0)

See Appendix A.

Decision letter (RSOS-181842.R1)

15-Mar-2019

Dear Dr Walker,

I am pleased to inform you that your manuscript entitled "Learning to live with interfering neighbours: The influence of time of learning and level of encoding on word learning" is now accepted for publication in Royal Society Open Science.

You can expect to receive a proof of your article in the near future. Please contact the editorial office (openscience_proofs@royalsociety.org and openscience@royalsociety.org) to let us know if

you are likely to be away from e-mail contact. Due to rapid publication and an extremely tight schedule, if comments are not received, your paper may experience a delay in publication.

on behalf of Professor Carolyn McGettigan (Associate Editor) and Professor Essi Viding (Subject Editor)
openscience@royalsociety.org

Associate Editor Comments to Author (Professor Carolyn McGettigan):
Thank you for your detailed response letter and revised manuscript. I am happy to accept the paper based on the changes you have made, without the need for further review.

Follow Royal Society Publishing on Twitter: [@RSocPublishing](https://twitter.com/RSocPublishing)
Follow Royal Society Publishing on Facebook:
<https://www.facebook.com/RoyalSocietyPublishing.FanPage/>
Read Royal Society Publishing's blog: <https://blogs.royalsociety.org/publishing/>

Appendix A

11th February 2019

Dear Professor McGettigan,

Re: Learning to live with interfering neighbours: The influence of time of learning and level of encoding

Thank you for the detailed feedback on the above paper. We have rewritten the paper in the light of these comments and feel that the manuscript is now substantially improved. We hope that you agree that we have been able to satisfactorily address the comments or provide justification where an alternative approach is taken, and that you feel the manuscript is now suitable for publication in *Royal Society Open Science*. Of course, we are very happy to respond to any further comments. The details of how we have addressed the specific issues raised are below.

I look forward to hearing from you in due course.

Yours sincerely,

Dr Sarah Walker

Reviewer #1

This paper reports on two experiments exploring factors affecting the consolidation of new vocabulary, and specifically the effects of exposure level and delay between learning and sleep. These are important questions and will be of interest to readers of RSOS. The experimental design and analyses seem generally sound (although I have one major comment, outlined below), and the paper is well written. The pattern of results across the two experiments is rather complicated, but the authors do a good job of interpreting them and placing them in the broader context of other research in the field.

I really have only one major concern that I think needs to be addressed, and that relates to the removal of quite a large amount of data due to non-compliance of participants with the instructions to avoid caffeine, alcohol and cigarettes. In the first experiment, the data of 20 participants were removed across Weeks 1 and 2, and in the second experiment 25/103 participants were excluded. I question this decision for a number of reasons. First of all, I wondered why such an exclusion might have been applied in the first place? If the participants were habitual smokers or caffeine or alcohol drinkers, might asking them to abstain not have interrupted their usual nocturnal sleep processes? I would have thought it would be better to have participants engage in their typical activities, and modify their behaviour as little as possible (obviously avoiding extreme behaviour such as binge drinking etc). I could not find any reference to a justification for these strict exclusion criteria, and it did not appear to be mentioned in the preregistration document for the first experiment.

Secondly, I would be concerned that the participants who had data removed may have differed from the included participants in other systematic ways that might be relevant to the present findings. For example, they may have been participants who were less committed to the experiment, and to the training required, and who therefore may have formed weaker lexical traces. Given that strength of initial encoding is one of the factors of interest in this study, it would seem important that this kind of individual variation within the population is captured.

At the very least, I think that a much stronger justification needs to be provided of the exclusion criteria, weighing the removal of

so much data up against the significant loss of power and the potential for the remaining data to be non-representative. I also think the data should be analysed with and without the excluded data. If there are no differences in the overall pattern of results, then the full set should be reported on (with a footnote explaining this). If there are differences, then both sets of analyses should be reported on, and the possible bases for any differences in results discussed.

We thank the reviewer for drawing our attention to this point. Exclusion criteria and restrictions relating to caffeine, smoking and alcohol are standard in this kind of research (e.g., Gais et al, 2011, JoCN), the less standard aspect is the online testing, which is known for more substantial data loss when response times are involved. In fact the number of datasets for which we have excluded but potentially usable data is very small. For example, in Experiment 1, only 6 participants were excluded from one or other week solely on the basis of failing to adhere to the restrictions (5 alcohol, 1 caffeine). These restrictions were in the pre-registration document (Data Collection Procedures), but we admit that it would have been useful to refer to this aspect in the Data Exclusion section as well. The majority of exclusions were partial (i.e., the data for the other week was retained) and were cases where either there were no data to be added (i.e., the participant did not take part in that test) or the data are unusable due to server errors or taking part outside the crucial requested time-window stipulated by condition. We were not aware that server errors would be a major source of data loss at the time of preregistration, but thought that it would be better to run sufficient participants to stick to the pre-registered participant number (80) than to run an experiment likely to be low on statistical power.

In the main, then, there is little data to add, and we disagree that it would be advantageous to analyse the dataset with these few participants included, given that the vast majority of participants adhered to the restrictions that were stipulated.

We have now clarified why such a large number of participants were excluded (page 8, paragraphs 1- 2).

Minor points:

- “Complementary Learning Systems” is sometimes capitalised

and sometimes not.

Fixed, thank you.

- p.6, line 12: “stronger acquisition of a motor adaption task”. I didn’t know what this meant.

We have now described this in more detail

- p. 13, line 5: “Participants were presented with the newly learned items mixed with foils”. What were the foils here, and how constructed?

The construction of the foils is explained in the stimuli section of the methods (page 9, paragraph 1). We have amended this sentence to make it clearer that the foils have already been described.

- Table 1 and 2 (and throughout): There are a very large number of tables, but many only include a few numbers and could easily be merged with other tables.

Thank you for this suggestion. The smaller tables (Tables 1 and 2, plus the equivalents for Experiment 2) have been merged. We did not see a good way to combine analysis summary tables without potentially confusing the reader, and so have left them as is.

- p. 21, line 40: Was Experiment 2 preregistered? Why Experiment 1 but not 2?

Experiment 2 was not pre-registered. We are relatively new to the pre-registration process and did not realise that follow-on studies could be pre-registered. That said, the similarity between the two experiments meant that preregistration of the new study was not so important, because the methods and analysis plans could be re-used.

Reviewer #2

This study investigates the conditions under which novel competitors to existing words induce lexical competition across a period including sleep. Two factors are manipulated: the time between learning and sleep, and the amount of exposure to the

novel words. Previously observed increases in lexical competition across sleep were replicated for standard and high exposure levels, but not for lower exposure levels. For the higher exposure levels, longer gaps between learning and sleep were associated with better explicit memory of novel words one week later.

A strength of this paper is its pre-registration, which makes transparent the predictions and experimental plans. The analysis strategy taken in the paper deviated from the pre-registration, however, which raises the question of whether the authors additionally ran the originally-planned analyses (if so, they should be reported).

The pre-registered maximal model analysis was run but did not converge, so we were unable to continue with this method (this is a common problem with maximal effects structures). As the use of mixed effects models is in its infancy, additional statistical training (and developments during the course of the study) helped us develop a better understanding of how to run mixed effects models, and therefore an alternative approach in line with Bates et al. (2018) was used.

Perhaps the biggest weakness of the paper is the lack of wake control groups, which limits the ability to make claims about the impact of sleep (as opposed to time passing). Though it would be a stronger paper with those groups, it is a solid enough contribution without them. This limitation deserves more explicit discussion, however.

We agree with the reviewer and have now included this as a limitation in the discussion (page 29, paragraph 3). There is evidence from Wang et al (2017) that suggests that a wake control condition would not show changes in lexical competition, but a direct comparison within this study would be more compelling.

The unpredicted relationship between delay and memory one week later with higher exposure levels appears to be weak and its mechanism is unclear; it seems likely to be spurious. I would encourage the authors to treat these results with even more caution than they are currently treated (for example, in the abstract, perhaps stating that “there was no clear relationship”

between delay and memory change, as opposed to taking seriously the complicated pattern of results).

We agree that we could better highlight the preliminary nature of this finding. As such, we have tempered our interpretation in the abstract “There was no evidence that going to bed soon after learning is advantageous for overnight consolidation; however, there was some preliminary suggestion that longer gaps between learning and bed-onset were associated with better explicit memory of novel words one week later, but only at higher levels of exposure.... Furthermore, the finding that longer-term explicit memory of stronger (but not weaker) traces benefit from periods of wake following learning deserves examination in future research.” As well as in the General Discussion... “Counter to previous claims [27-29], there was no evidence that less time awake between learning and sleep was beneficial for the overnight consolidation of novel orthographic forms in the present adult sample. Bolstering this conclusion, there was some tentative evidence that better recall and recognition of the novel words one week after training was associated with *more* time awake between training and sleep. Although requiring replication and further investigation, this finding possibly relates to...”

It was very difficult to flip back and forth to the tables, figures, and figure legends; embedded figures and tables would facilitate review.

We agree with the reviewer that this would make the review process easier, but unfortunately the journal template asks us to format the document as it stands.

It is not clear that the analysis code is / will be made available.

The link to the analysis script is provided on page 32 under Data Accessibility.

Reviewer #3

The study uses a really nice sample size, but what was the motivation for testing 113 participants? The manuscript states the preregistered target was 80, it might be worth mentioning a bit more about where that target came from, and make it clearer

why data collection wasn't stopped when this number was reached.

We thank the reviewer for this point. The participants section was poorly worded, but is hopefully clearer now, given the response to Reviewer 1. Our target was 80 usable datasets, and the various reasons for loss of online data meant that we needed to test 113 participant in all. We could not feasibly stop at *exactly* 80 participants because it was hard to ensure that counterbalancing would operate effectively given that the usability of a participant's data might not be known until at least two weeks after scheduling. Therefore we decided to modestly overshoot our target with the view that a few too many is better than a slight undershoot.

It's not entirely clear that the study was completed online via qualtrics until around page 10. This should be clarified, and perhaps raised earlier on as one of the contributions of this literature to the field – many of the prior studies of sleep/lexical integration have involved laboratory-style tests – the value of implementing this online should be highlighted.

We have clarified the online nature of the study earlier in the manuscript (page 7, paragraph 1), and highlighted its value as a naturalistic procedure.

Some more detailed motivation for including the semantic categorisation task is needed (p7 par 1), and perhaps earlier on too. How/why is it a marker of lexical integration? You could discuss this in more detail on p4 par 2. This paragraph needs some restructuring to make it clearer how the semantic categorisation task is a marker of lexical integration.

This has now been added to page 4, paragraph 2 of the revised manuscript. We thank the reviewer for drawing our attention to this.

The reversal of responses on the speeded recognition task is a bit concerning (while I'm sympathetic to the logic). It might be safer to avoid this. My recommendation is to exclude these participants' data from analysis if they are deemed to be outliers, since we can only be so confident in this assumption. Given the great sample size here, there should be sufficient power to do this and still have a good amount of data left over.

This could be done, but would in our view make no difference to what turned out to be an uninformative analysis. More importantly, we think that our description of the procedure in our initial submission was too vague and so may have led the reviewer to the view that there was risk of data contamination in our procedure. More detail is provided here.

In Experiment 1, where this problem was more common, it affected 14/342 test sessions (4%). The distribution of session accuracy for this experiment is given below, showing the clustering of the excluded sessions close to zero accuracy. This distribution nicely mirrors the distribution of the included datasets, with zero accuracy (perfect inaccuracy) being the modal performance, just as perfect accuracy is the modal performance for the included items. These values are all significantly below chance performance (for the least extreme excluded dataset, $p = .00009$; for the majority of the excluded datasets this p -value drops below $.00000001$), and so we can be fully confident that the participants were not responding randomly. We see no other viable interpretation of this level of performance. We have added more detail to the section describing this procedure, which we hope will reassure the reviewer.

I am not familiar with using $p < .2$ as a threshold for justification of fixed/random effects in a model. Can you give readers more detail about how you came to this decision?

We thank the reviewer for this point. On page 14, paragraph 3, we now cite Barr et al. (2013), who recommend the use of $p < .2$ in order to guard against anti-conservativity.

Page 10 – 11; how certain can we be about participants' adherence to these instructions – I imagine you checked the test time for each participant to make sure they stuck to the schedule? This kind of detail should be reported. Similarly, did you collect information on self-reported sleep duration in between day 1 and day 2? Was this similar across the different groups?

More detail has now been provided in the methods about participant's adherence to the instructions, this can be found on page 8, paragraph 2 and page 10, paragraph 2. We did not collect self-reported sleep duration. It would have been somewhat useful to ensure that the groups were well matched,

although this variable tends not to be a useful predictor of memory consolidation.

Minor comments

Learning to live with interfering neighbours: The influence of time of learning and level of Encoding.. Title – the last part feels incomplete? On what - lexical integration?

Page 2 line 13: "Participants made speeded semantic decisions (i.e., natural/artefact) to the existing words...." – ? this is unclear

Page 4 line 28: "Bowers et al. trained novel word forms..." – incomplete sentence? This section needs reworking.

Page 4 line 38: response times were slower to the words... – slower for what? Was this for accurate responses only?

Page 7 line 19: clarify which kind of online you mean here

Page 7 line 33: a more intensive level... of exposure?

Incomplete sentence

Page 14 line 47: there is a rogue bracket

We thank the reviewer for spotting these. The title has now been amended to: *Learning to live with interfering neighbours: The influence of time of learning and level of encoding on word learning*. The other points have all been attended to.